# A Mutually Exciting Latent Space Hawkes Process Model for Continuous-time Networks

**Zhipeng Huang**[1]      **Hadeel Soliman**[1]      **Subhadeep Paul**[2]      **Kevin S. Xu**[1]

[1]Department of Electrical Engineering and Computer Science, University of Toledo, Toledo, OH, USA
[2]Department of Statistics, The Ohio State University, Columbus, OH, USA

## Abstract

Networks and temporal point processes serve as fundamental building blocks for modeling complex dynamic relational data in various domains. We propose the *latent space Hawkes (LSH)* model, a novel generative model for continuous-time networks of relational events, using a latent space representation for nodes. We model relational events between nodes using mutually exciting Hawkes processes with baseline intensities dependent upon the distances between the nodes in the latent space and sender and receiver specific effects. We demonstrate that our proposed LSH model can replicate many features observed in real temporal networks including reciprocity and transitivity, while also achieving superior prediction accuracy and providing more interpretable fits than existing models.

## 1 INTRODUCTION

Dynamic networks are used to represent time-varying relationships (edges) between a set of nodes. They are useful in a variety of application settings, including messages between users on online social networks and transactions between users on online marketplaces. In such settings, the network typically evolves over time through a set of *timestamped relational events*. Each event is a triplet $(u, v, t)$ denoting that node $u$ initiated an interaction with node $v$ (e.g. $u$ sent a message to $v$) at timestamp $t$. We refer to this type of dynamic network as a *continuous-time network* because it is continuously evolving through these relational events.

A topic of much recent interest is identifying latent representations for nodes in networks. These latent representations are often referred to as node embeddings, and node embedding-based approaches for common network analysis tasks including link prediction have gained significant attention in recent years [Grover and Leskovec, 2016, Goyal

and Ferrara, 2018, Cui et al., 2018]. Prior to this surge of interest, latent space models have been used in statistics and mathematical sociology for exploratory analysis of networks [Hoff et al., 2002, Hoff, 2005, 2007, Handcock et al., 2007, Krivitsky et al., 2009]. Latent representations have also been developed for dynamic networks evolving over discrete time steps [Sewell and Chen, 2015] or in continuous time [Nguyen et al., 2018].

Latent space representations can be combined with temporal point processes (TPPs) to form a probabilistic generative model for continuous-time networks, which we consider in this paper. Augmenting the latent representation with a TPP enables one to generate timestamps for the edges between nodes. Yang et al. [2017] proposed the dual latent space (DLS) generative model that combines two types of latent spaces with bivariate Hawkes processes. They found that using two types of latent spaces, one to capture homophily and one to capture reciprocity, provides a richer model that also leads to improved link prediction accuracy. However, much of the interpretability of the latent space, which was the original motivation of the latent space model of Hoff et al. [2002], is lost by using multiple high-dimensional latent spaces. Furthermore, the DLS model has issues with stability of the generative process due to the multiple latent spaces. It also uses only reciprocal excitation and not self excitation. Self excitation is important in application settings such as modeling text messages, where person $u$ may send multiple messages to $v$ in rapid succession before $v$ responds.

In this paper, we consider using a single latent space representation to provide a more interpretable model. The single latent space limits the flexibility of the model compared to the DLS, so we increase flexibility by adding self excitation and sender and receiver effects. We demonstrate that our proposed latent space Hawkes (LSH) model is competitive with other models in predictive and generative tasks on 4 real network datasets while providing more interpretable and stable model fits. Furthermore, we apply our LSH model to perform exploratory analysis on a dataset of militarized disputes to reveal network structure between countries.

*Accepted for the 38th Conference on Uncertainty in Artificial Intelligence* (UAI 2022).

# 2 BACKGROUND

## 2.1 HAWKES PROCESSES

The Hawkes process model was introduced for temporal point processes by Hawkes [1971]. The defining characteristic of a Hawkes process is that it is self exciting, meaning that each event increases the rate of future events for some period of time. Mutually exciting Hawkes processes allow events from different processes to excite each other in addition to self excitation [Laub et al., 2021]. An $m$-dimensional mutually exciting Hawkes process is characterized by a conditional intensity function for each dimension $i$:

$$\lambda_i^*(t) = \lambda_i(t|\mathcal{H}_t) = \mu_i + \sum_{j=1}^{m} \sum_{k:t_k<t} \phi_{ij}(t - t_k), \quad (1)$$

where $\mathcal{H}_t$ denotes the history of the process up to time $t$, $\mu_i$ denotes the baseline rate of events in dimension $i$, and $\phi_{ij}(\cdot)$ is a kernel function that describes how an event in dimension $j$ influences dimension $i$.

The most commonly used kernel function is the exponential kernel $\phi(t - t_k) = \alpha\beta e^{-\beta(t-t_k)}$ for $\alpha > 0$ and $\beta > 0$. With each event arrival, the conditional intensity jumps by $\alpha$. The influence of the arrival then exponentially decays at rate $\beta$ over time. In practice, both $\alpha$ and $\beta$ are unknown parameters that need to be estimated from data, which is usually done using maximum likelihood estimation [Laub et al., 2021]. However, estimators for the decay parameter $\beta$ are poorly behaved [Santos et al., 2021], and it is more computationally efficient to choose a fixed $\beta$ rather than estimating it [Lemonnier and Vayatis, 2014].

An approach that is more general than fixing the value of $\beta$ is the sum of exponential kernels method [Lemonnier and Vayatis, 2014], which defines $\phi(t - t_k) = \sum_b^B \alpha\beta_b e^{-\beta_b(t-t_k)}$, where $B$ denotes the number of exponential kernels. This method generalizes better as it handles different time scales, which makes the modeling less sensitive to choice of $\beta$. We use the sum of exponential kernels decay in this paper.

## 2.2 LATENT SPACE MODELS

The latent space model (LSM), first proposed by Hoff et al. [2002] is a popular model-based approach for social network analysis. Designed initially for a single static undirected network, the LSM allows the probability of an edge between two nodes to depend on their Euclidean distance in an unobserved or latent space using a logistic regression model. Let $A$ denote the adjacency matrix of a network, with $a_{uv} = 1$ for node pairs $(u, v)$ with an edge and $a_{uv} = 0$ otherwise. By assuming conditional independence between node pairs, the log-likelihood can be written as

$$\log P(A|\eta) = \sum_{u<v} [\eta_{uv}a_{uv} - \log(1 + e^{\eta_{uv}})],$$

where entry $\eta_{uv}$ in the matrix $\eta$ denotes the log odds of an edge being formed between nodes $(u, v)$. $\eta_{uv}$ is parameterized as follows: $\eta_{uv} = \xi - \|z_u - z_v\|_2$, where $z_u$ denotes the latent position of node $u$ in a $d$-dimensional latent space, and $\xi$ is an intercept term. Under this parameterization, two nodes with closer latent positions have higher probability of forming an edge.

The latent space model provides a visual and interpretable model-based spatial representation of social relationships. It has been extended by many researchers. Handcock et al. [2007] developed a latent position cluster model to capture transitivity, homophily, and community structure simultaneously. The latent space models were later extended to include node-specific random effects by Krivitsky et al. [2009]. Latent space models have also been extended for more complex network based data structures, including multiple networks [Gollini and Murphy, 2016, Salter-Townshend and McCormick, 2017], discrete-time dynamic networks [Sewell and Chen, 2015, 2016, Friel et al., 2016, Gracious et al., 2021], and multimodal networks [Wang et al., 2019]. We use the latent space model as the building block for our proposed continuous-time LSH model.

## 2.3 RELATED WORK

**Dynamic Network Embeddings**  One line of related work is focused on node embeddings for dynamic networks. Compared to static network embedding methods, dynamic network embedding methods assign nodes low-dimensional representations that effectively preserve the temporal information. Nguyen et al. [2018] proposed continuous-time dynamic network embeddings (CTDNE), a general framework to learn a time-respecting embedding from continuous-time dynamic networks. Their framework acts as a basis for incorporating temporal dependencies into existing node embedding and deep graph models based on random walks. Other approaches for dynamic network embedding have also been proposed [Chen et al., 2018, Sankar et al., 2018, Goyal et al., 2020], many of which are discussed in a recent survey on dynamic network embedding [Xie et al., 2020].

**TPP-based Network Models**  TPP-based network models are generative models for continuous-time dynamic networks that incorporate both a generative process for the nodes $(u, v)$ that form an edge and the time $t$ at which an edge is formed. These timestamped edges or events can be viewed as triplets $(u, v, t)$. Many TPP-based network models utilize a discrete latent variable representation for the nodes [Blundell et al., 2012, DuBois et al., 2013, Miscouridou et al., 2018, Junuthula et al., 2019, Arastuie et al., 2020, Soliman et al., 2022], dividing them into different blocks or communities.

The most closely related work to this paper is the dual latent space (DLS) model [Yang et al., 2017], which also utilizes

a continuous latent variable representation inspired by the latent space model. The DLS model uses bivariate Hawkes processes to capture the homophily and reciprocity of dynamic networks. They observed that the latent dimensions of users which affect link formation may be different from the latent dimensions of users which affect reciprocity. We discuss shortcomings of the DLS model and its relation to our proposed model in Section 3.2.

Another TPP-based network model using a continuous latent space is proposed by Rastelli and Corneli [2021]. It assumes that the latent positions of nodes may change at a set of predefined change points rather than being fixed over time.

**Other Continuous-time Network Models** Earlier research on continuous-time network models was proposed by Wasserman [1980a,b], who modeled the evolution of network data using continuous-time Markov chains. Later on, Snijders [2005], Snijders et al. [2017] proposed a set of network models that offers more flexibility to represent a variety of network effects, such as transitivity, reciprocity, etc. Fan and Shelton [2009] explored the inference for these models and proposed a sampling-based learning algorithm for continuous-time social network models.

# 3  PROPOSED MODEL

In our model, we employ a latent space to learn hidden node attributes underlying the network and mutually exciting Hawkes processes to capture the temporal dynamics of communication. We model the communications between each pair of nodes as realizations from a bivariate Hawkes process whose conditional intensity function $\lambda_{uv}(t|\mathcal{H}_t)$ includes three components: a baseline rate, a self-exciting term, and a reciprocal term.

Let $z_u$ and $z_v$ denote the latent positions for nodes $u$ and $v$, respectively. We model baseline rate $\mu_{uv}$ as a function of Euclidean distances between $z_u$ and $z_v$. Gollini and Murphy [2016] showed that squared Euclidean distances are computationally more efficient than Euclidean distances yet resulted in similar latent positions. Thus, we use squared Euclidean distances $||z_u - z_v||_2^2$ in the model for $\mu_{uv}$, similar to DLS [Yang et al., 2017]. We further add sender and receiver node effect terms $\delta_u, \gamma_v$ to the model as in Hoff [2005], Krivitsky et al. [2009], Wang et al. [2019] to capture the degree heterogeneity, namely the tendency of some nodes to send and receive events more than others, respectively.

A Hawkes process with exponential kernel has been found to be a good model for conversation event sequences as well as other relational temporal event data [Masuda et al., 2013]. We use a sum of $B$ exponential kernels in our Hawkes processes. We set $\beta = (\beta_1, \beta_2, \ldots, \beta_B)$ as a set of fixed known decays and $C = (C_1, C_2, \ldots, C_B)$ as a set of scaling parameters for the kernel with $\sum_i^B C_i = 1$. The conditional

intensity function can be written as follows:

$$\lambda_{uv}^*(t) = \mu_{uv} + \sum_{t_{uv} < t} \sum_b^B C_b \alpha_1 \beta_b e^{-\beta_b(t-t_{uv})}$$
$$+ \sum_{t_{vu} < t} \sum_b^B C_b \alpha_2 \beta_b e^{-\beta_b(t-t_{vu})}, \quad \forall u \neq v \tag{2}$$

where the baseline rate $\mu_{uv}$ is given by

$$\mu_{uv} = e^{-\theta_1 ||z_u - z_v||_2^2 + \theta_2 + \delta_u + \gamma_v}. \tag{3}$$

## 3.1  MODEL PARAMETERS

The LSH model has parameters $(Z, \alpha_1, \alpha_2, \theta_1, \theta_2, \delta, \gamma)$. Each node has a $d$-dimensional latent position $z_u$, a sender propensity parameter $\delta_u$ and a receiver propensity parameter $\gamma_u$. $\alpha_1$ and $\alpha_2$ are the jump size parameters for self-excitation and reciprocal-excitation. $Z$ is a $n \times d$ matrix where each row is a latent position vector $z_u$ for a node, and $d$ is the latent dimension. Each of $\delta$ and $\gamma$ is a vector of size $n$. $\theta_1$ and $\theta_2$ are slope and intercept terms, respectively, associated with the baseline rate and latent positions. A positive slope $\theta_1$ provides node pairs closer together in the latent space with a higher probability of forming edges, while a negative slope does the reverse.

**Identifiability** There are two sets of identifiability problems that need to be discussed. From the observed event times, the Hawkes process parameters $\mu_{uv}, \alpha_1, \alpha_2$ can be identified as shown by Ozaki [1979]. With the baseline intensity parameter $\mu_{uv}$ correctly identified, we explore the identifiability of the parameters in the model for $\mu_{uv}$. The identifiability of parameters in the latent space model has been discussed by Ma et al. [2020] for a single network and by Zhang et al. [2020] for multilayer networks.

Denote $1_n$ to be the $n$ dimensional vector and $J_n = 1_n 1_n^T$ to be the $n \times n$ matrix whose elements are all 1's. We first note that the magnitude of the parameter $\theta_1$ is not identifiable since it enters the equation for $\mu$ as a product with $||z_u - z_v||_2^2$. However, the sign of $\theta_1$ is identifiable since $||z_u - z_v||_2^2$ is always positive. In the following, we set $\theta_1 = 1$ and examine the conditions for identification of other parameters. We have

$$\log(\mu_{uv}) = \theta_2 - ||z_u||^2 - ||z_v||^2 + z_u^T z_v + \delta_u + \gamma_v$$
$$= \theta_2 + z_u^T z_v + \tilde{\delta}_u + \tilde{\gamma}_v,$$

where $\tilde{\delta}_u = \delta_u - ||z_u||^2$ and $\tilde{\gamma}_v = \gamma_v - ||z_v||^2$. Now let $\tilde{\delta}$ and $\tilde{\gamma}$ denote the $n$-dimensional vectors whose elements are $\tilde{\delta}_u$ and $\tilde{\gamma}_v$, respectively. (All vectors are column vectors.) Writing in matrix form, the above expression is

$$\log(\mu) = \theta_2 J_n + Z Z^T + \tilde{\delta} 1_n^T + 1_n \tilde{\gamma}^T.$$

**Theorem 3.1.** *Under the following assumptions:*

1. *The latent positions are centered, i.e., $HZ = Z$, where $H = I - \frac{1}{n}11^T$, and*

2. *The total nodal effects sum to 0, i.e., $1_n^T \tilde{\delta} = 0$ and $\tilde{\gamma}^T 1_n = 0$,*

*if two sets of parameters $\theta_2, Z, \gamma, \delta$ and $\theta_2', Z', \gamma', \delta'$ lead to the same $\log(\mu)$, then*

$$\theta_2 = \theta_2', \ \delta = \delta', \ \gamma = \gamma' \ \text{and} \ Z = Z'O,$$

*where $O$ is a $d \times d$ orthogonal matrix.*

The proof is provided in Appendix A.1. Thus, under the constraints that the true latent positions $Z$ are centered and total nodal sender and receiver effects sum to 0, the parameters $\theta_2, \delta, \gamma$ and the vector distances $ZZ^T$ are exactly identified, while $Z$ is identified up to an orthogonal matrix $O$.

## 3.2 RELATION TO DLS MODEL

The most similar model to ours is the dual latent space (DLS) model [Yang et al., 2017]. It uses the following form for the conditional intensity function[1]:

$$\lambda_{uv}^*(t) = e^{-||z_u - z_v||_2^2 + \theta_2}$$
$$+ \sum_{t_{vu} < t} \sum_{b}^{B} \alpha_2 e^{-||x_u^{(b)} - x_v^{(b)}||_2^2} \beta_b e^{-\beta_b(t - t_{vu})}, \quad \forall u \neq v \quad (4)$$

By comparing the form of the conditional intensity function for DLS (4) with that of our proposed LSH model (2), we identify 3 key differences, each addressing a concern regarding the DLS model:

1. The DLS utilizes reciprocal latent spaces $X^{(b)}$ to allow different rates of reciprocity between node pairs. This increase in flexibility of the model comes with a key drawback: the estimated latent positions for a node pair $(u, v)$ and kernel $b$ may result in the jump size $\alpha_2 e^{-||x_u^{(b)} - x_v^{(b)}||_2^2} > 1$, which leads to an unstable process. We were unable to simulate new networks from the DLS model fits to real networks due to the instability as we discuss in Section 5.2.2. In contrast, we use just a single jump size $\alpha_2$ for all node pairs in our LSH model. While this may be less flexible, it does not lead to instability like the reciprocal latent space.

2. The DLS does not have a self excitation component, unlike our proposed LSH (second term in (2)). The lack of self excitation prevents the DLS from modeling repeated edges from node $u$ to $v$ with no response from $v$ back to $u$. For example, this setting occurs frequently in militarized conflicts between countries, where one country repeatedly threatens or takes action against another country that does not retaliate.

3. The DLS does not have nodal effects parameters ($\delta_u$ and $\gamma_v$ in (2)). This limits its ability to model nodes with different rates of sending or receiving events.

Furthermore, a primary motivation of the latent space model is to embed the network into a single Euclidean space that can be easily visualized and interpreted. By using a single latent space, our proposed LSH is able to provide a much more interpretable model fit compared to DLS.

## 4 ESTIMATION PROCEDURE

Our model consists of mutually exciting bivariate Hawkes processes over all pairs of nodes. Using the likelihood theorem of Daley and Vere-Jones [2003], we can write the log-likelihood as

$$\log \mathcal{L} = \sum_{u \neq v} \left\{ \sum_{i=1}^{k} \log(\lambda_{uv}^*(t_i)) - \int_0^{t_k} \lambda_{uv}^*(t)dt \right\}, \quad (5)$$

where $k$ denotes the total number of events and $\lambda_{uv}^*(t)$ takes on the form in (2). We simplify the log-likelihood and improve the efficiency of the estimation by deriving a recursive form as in Ozaki [1979]. More details and the full log-likelihood derivation for our LSH model are provided in Appendix A.2, resulting in the simplified expression in (A.4).

Latent space models typically assume that the probability of forming an edge between two nodes is inversely proportional to the distances between the node positions in the latent space. Thus, the observation of an edge between two nodes typically pulls them closer together in the latent space. The presence of the slope parameter $\theta_1$ in the baseline rate $\mu_{uv}$ for our LSH model (3) allows us to either pull node pairs with events closer together by constraining $\theta_1 > 0$ or push them further apart by constraining $\theta_1 < 0$. Or, we could leave $\theta_1$ unconstrained—we find that this usually results in the estimate $\hat{\theta}_1 > 0$.

We use the L-BFGS-B algorithm [Byrd et al., 1995] to minimize the negative log-likelihood (NLL). The gradients of the log-likelihood can be carried out using the Autograd package [Maclaurin et al., 2015] for automatic differentiation of standard Python functions. We consider also an alternating minimization approach that alternates between estimating the latent space and the model parameters, which we show in Appendix A.3. Our alternating minimization approach is partly inspired by the projected gradient method of Ma et al. [2020], which also alternates between estimating the latent space and the model parameters in a static latent space model. We find that the alternating minimization approach generally converges more slowly than L-BFGS-B, so the results we present in this paper use L-BFGS-B.

We use a multidimensional scaling algorithm as an initialization for the latent space positions $Z$ as in the orig-

---

[1]They include also a periodic kernel in addition to the exponential kernels, which we exclude for ease of comparison.

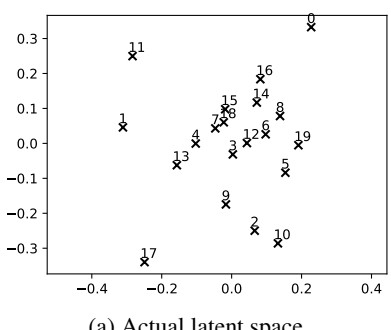

(a) Actual latent space

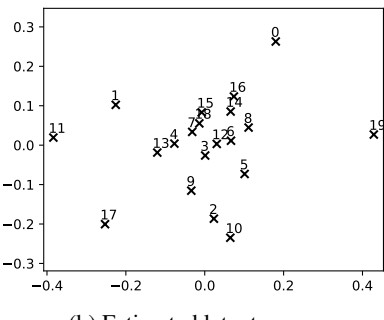

(b) Estimated latent space

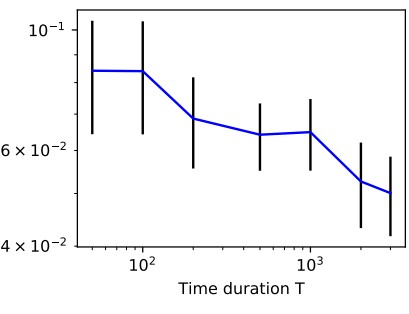

(c) Latent positions estimation error

Figure 1: Comparison of (a) actual latent space and (b) estimated latent space (with Procrustes transformation) on a 20 node simulated network with duration $T = 100$. The recovered latent node positions are close to the actual positions. (c) The RMSE over 30 simulated networks ($\pm 2$ standard errors) decreases as the duration $T$ increases.

inal latent space model proposed by Hoff et al. [2002]. We set random values to initialize all other parameters $\Theta = (\alpha_1, \alpha_2, \theta_1, \theta_2, \delta, \gamma)$.

## 5 EXPERIMENTS

In this section, we perform evaluation tasks for our proposed model on simulated networks and real networks[2]. We use a sum of $B = 3$ exponential kernels and utilize decays with time scales of an hour, a day, and a week, which is the same as Yang et al. [2017] did in their DLS model. We also fix $C = (1/3, 1/3, 1/3)$ for simplicity[3].

### 5.1 SIMULATED NETWORKS

We first test our L-BFGS-B estimation procedure on networks simulated from our latent space Hawkes (LSH) model. We simulate networks of 20 nodes in a 2-D latent space using parameters $\theta_1 = 1$, $\theta_2 = -3.2$, $\alpha_1 = 0.01$, and $\alpha_2 = 0.02$. Each dimension of the latent positions as well as sender and receiver effects for nodes are sampled independently from a standard Normal distribution: $z_u, \delta_u, \gamma_u \sim \mathcal{N}(0, 1)$. We increase the time duration $T$ from 50 to $3,000$ and evaluate the estimation accuracy for the latent positions and other parameters. Additional details on the simulation process is provided in Appendix B.1. A comparison of the actual and estimated latent positions for a simulated network is shown in Figure 1 along with the root mean squared error (RMSE) for estimated latent positions over 30 simulated networks. As expected, the error decreases for increasing time duration $T$. The error for the other parameters decreases also, as we show in Figure B.1 in Appendix B.1. Thus, L-BFGS-B appears to accurately estimate latent positions and model parameters.

Table 1: Summary statistics of real network datasets

| Dataset | Nodes | Events | Time Duration |
|---------|-------|--------|---------------|
| Reality | 65 | $2,150$ | 8 months |
| Enron | 155 | $9,646$ | 15 months |
| MID | 145 | $5,088$ | 23 years |
| FB-forum | 899 | $33,720$ | 5.5 months |

### 5.2 REAL NETWORKS

We perform experiments on several real network datasets: Reality Mining [Eagle and Pentland, 2006], Enron emails [Klimt and Yang, 2004], Militarized Interstate Disputes (MID) [Palmer et al., 2021], and Facebook-forum [Rossi and Ahmed, 2015]. Summary statistics for the datasets are shown in Table 1, and additional details are provided in Appendix B.2. Each dataset consists of a set of relational events, each denoted by a sender, a receiver, and a timestamp.

**Baselines for Comparisons** We compare against three other Hawkes process-based continuous-time network models. The dual latent space (DLS) model [Yang et al., 2017] is the most similar to ours, and we provide a detailed comparison of the DLS model with our proposed LSH model in Section 3.2. We also compare against two recently proposed Hawkes process-based block models: the community Hawkes independent pairs (CHIP) model [Arastuie et al., 2020] and the block Hawkes model (BHM) [Junuthula et al., 2019]. Finally, we compare also against the non-generative continuous-time dynamic network embeddings (CTDNE) [Nguyen et al., 2018] approach. Additional information on these models for comparison along with implementation details are provided in Appendix B.3.

### 5.2.1 Predictive Accuracy

We first evaluate the predictive ability of our proposed LSH model. We split each dataset into a training set containing

---

[2]Python code to reproduce our results is available at `https://github.com/IdeasLabUT/Latent-Space-Hawkes`

[3]We also experimented with estimating $C$ but did not find much difference in the results.

Table 2: Evaluation metrics for predictive accuracy on real network datasets. Bold entry denotes highest accuracy for each metric on a dataset. Test log-lik. shows the mean test set log-likelihood per event and the number of latent dimensions $d$ or blocks $K$ that maximize it. The AUC column shows the mean (standard deviation) of the AUC across 100 time points for dynamic link prediction. DLS does not scale to the FB-forum data. CTDNE is not generative so test log-likelihood is not applicable.

| Dataset | Model | Test log-lik. | AUC |
|---------|-------|---------------|-----|
| Reality | LSH | $\mathbf{-3.71}\,(d=4)$ | $0.945(0.028)$ |
|         | DLS | $-5.64\,(d=300)$ | $0.940(0.034)$ |
|         | BHM | $-5.31\,(K=50)$ | $\mathbf{0.957(0.022)}$ |
|         | CHIP | $-4.70\,(K=1)$ | $0.937(0.028)$ |
|         | CTDNE | | $0.936(0.033)$ |
| Enron | LSH | $\mathbf{-4.87}\,(d=4)$ | $0.946(0.024)$ |
|       | DLS | $-5.29\,(d=100)$ | $\mathbf{0.947(0.017)}$ |
|       | BHM | $-6.35\,(K=14)$ | $0.839(0.035)$ |
|       | CHIP | $-5.34\,(K=4)$ | $0.895(0.053)$ |
|       | CTDNE | | $0.912(0.035)$ |
| MID | LSH | $\mathbf{-3.38}\,(d=3)$ | $\mathbf{0.988(0.018)}$ |
|     | DLS | $-4.52\,(d=100)$ | $0.977(0.007)$ |
|     | BHM | $-4.97\,(K=95)$ | $0.971(0.031)$ |
|     | CHIP | $-3.63\,(K=2)$ | $0.958(0.035)$ |
|     | CTDNE | | $0.953(0.018)$ |
| FB-forum | LSH | $\mathbf{-7.21}\,(d=8)$ | $\mathbf{0.932(0.009)}$ |
|          | BHM | $-11.16\,(K=57)$ | $0.839(0.017)$ |
|          | CHIP | $-7.65\,(K=2)$ | $0.919(0.011)$ |
|          | CTDNE | | $0.788(0.028)$ |

the first $80\%$ of events and a test set containing the remaining $20\%$ of events. We estimate model parameters on the training set and evaluate prediction accuracy on the test set. We choose the number of latent dimensions $d$ (for LSH and DLS) and the number of blocks $K$ (for BHM and CHIP) that maximizes the log-likelihood evaluated on the test set.

**Test Log-likelihood** We use the mean log-likelihood per event on the test set, also used by DuBois et al. [2013] and Arastuie et al. [2020], as an evaluation metric for the model's predictive ability on future data. As shown in Table 2, our Latent Space Hawkes (LSH) significantly outperforms the other models on all datasets. The test log-likelihood is maximized for the LSH at relatively small latent dimensions $d$ compared to the DLS model. The low-dimensional latent representation using a single latent space makes the LSH fit more interpretable than the high-dimensional DLS representation using multiple latent spaces. Furthermore, these results suggest that the addition of nodal effects and self excitation in the LSH significantly affects the predictive ability compared to DLS.

**Dynamic Link Prediction** We further explore the performance of the learned model in a dynamic link prediction task. We use the same experiment set-up as Yang et al. [2017]. We randomly sample 100 time points $t_i$ during the test period. We then compute the probability of a link appearing between each pair of nodes in the $[t_i, t_i + \delta]$ time window. We set $\delta$ to be two weeks for the Reality, Enron, and FB-forum datasets and two months for the MID data, which takes place over a longer period of time. For each of these $\delta$ intervals, we obtain the Receiver Operating Characteristics (ROC) curve and compute the Area Under the Curve (AUC) measured across all pairs of nodes according to the predicted probabilities given by the model.

The mean AUC values are shown in Table 2 with the value inside the parentheses indicating the standard deviation over these 100 time intervals. The ROC curves and box plots for the corresponding AUC values are presented in Appendix B.4. Our proposed LSH model is competitive at the dynamic link prediction task, achieving highest mean AUC on FB-forum and MID and second highest on Reality and Enron.

### 5.2.2 Generative Accuracy

To evaluate generative accuracy of our proposed LSH model, we simulate networks with our fitted parameters and perform posterior predictive checks (PPCs) using network statistics such as reciprocity and transitivity. While our LSH model has no issues simulating networks, the DLS is problematic due to its model formulation. The jump size for reciprocal excitation depends on distances between nodes in a reciprocal latent space and is further scaled by the parameter $\alpha_2$ in (4). Since the maximum jump size is not constrained, this results in some node pairs having unstable Hawkes processes so that the simulation does not terminate. To enable us to make comparisons with the DLS model, we stabilize it by fixing the scaling parameter for the jump size $\alpha_2 = 1$.

We simulate 15 networks from the fitted model on each real dataset, with the exception of DLS, which does not scale to the FB-forum data. We then perform PPCs on the number of events generated, average run length, and 4 static network statistics: transitivity (global clustering coefficient), reciprocity, average local clustering coefficient (LCC), and average degree. The run length is the number of consecutive events in the same direction, e.g. in the sequence $(u, v), (v, u), (v, u), (v, u), (v, u), (u, v)$, the run length for $(v, u)$ is 4 because it appears 4 times consecutively before the reciprocal event $(u, v)$ appears.

A comparison between the actual statistics and mean simulated statistics is shown in Table 3. We compare LSH and DLS since they are both based on the latent space model. The DLS model generates significantly more events than exist in the actual network, ranging from roughly a 4x increase (Reality) to an 80x increase (MID). We believe that this is

Table 3: Comparison of generative accuracy between models using mean statistic over 15 simulated networks. Bold entry denotes the simulated statistic closest to the actual statistic. While both LSH and DLS can replicate the static network statistics from the actual networks, DLS generates way too many events compared to the actual networks.

| Dataset | Statistic | Actual | LSH | DLS |
|---------|-----------|--------|-----|-----|
| Reality | # of events | 2,148 | **2,190** | 9,493 |
| | Avg. run length | 2.49 | **2.62** | 1.91 |
| | Transitivity | 0.29 | 0.34 | **0.32** |
| | Reciprocity | 0.80 | **0.86** | 0.52 |
| | Avg. LCC | 0.25 | 0.19 | **0.21** |
| | Avg. degree | 4.86 | **4.45** | 7.50 |
| Enron | # of events | 9,646 | **11,010** | 675,621 |
| | Avg. run length | 2.44 | **2.63** | 1.87 |
| | Transitivity | 0.31 | 0.39 | **0.30** |
| | Reciprocity | 0.65 | **0.65** | **0.65** |
| | Avg. LCC | 0.40 | 0.51 | **0.36** |
| | Avg. degree | 18.46 | 25.86 | **18.43** |
| MID | # of events | 5,088 | **3,996** | 412,890 |
| | Avg. run length | 2.88 | **2.71** | 1.89 |
| | Transitivity | 0.13 | 0.24 | **0.20** |
| | Reciprocity | 0.64 | **0.57** | 0.52 |
| | Avg. coef | 0.25 | **0.29** | **0.29** |
| | Avg. degree | 6.80 | **7.05** | 9.57 |

due to the reciprocal latent space used in the DLS model. Even though we stabilized the model by setting $\alpha_2 = 1$, some nodes are likely still extremely close in the reciprocal latent space, causing too many events to be generated.

We also find that the lack of self-excitation in DLS prevents it from replicating the run length of directed event sequences. Since DLS only has reciprocal excitation, its generated networks have the average run length of about 2 regardless of the average run length in the actual network. On the other hand, the DLS model performs quite well at replicating the static network statistics, and in many cases, even better than our proposed LSH. We believe that this is partially due to the much higher latent dimension $d$ that maximizes the test log-likelihood for DLS. The LSH could potentially achieve better generative accuracy using higher $d$ as well. Additional results on generative accuracy, including plots comparing the actual statistics with the distribution of the simulated statistics, are provided in Appendix B.5.

## 6   CASE STUDY

We now present a case study demonstrating our proposed LSH model being used for exploratory analysis on a real continuous-time network: the Militarized Interstate Disputes (MID) incident network. Timestamped edges in this network correspond to individual incidents within disputes between countries. Incidents include threats, displays, and uses of force initiated by one country towards another.

Incidents in the MID network are indicative of negative relationships between countries. As a result, one might expect the network to be disassortative. On the other hand, incidents frequently occur between countries that are geographically close, particularly if they share a boundary, which suggests that the network may also have an assortative structure. Thus, we conduct exploratory analysis of this network using two different parameterizations of our model. We fix the latent dimension to be $d = 2$ in both models so that we can visualize the latent positions of the countries.

We first consider a *positive slope* model by constraining $\theta_1 > 0$ in (3) so that two countries with lots of incidents between them are pulled closer together in the latent space, as is typically the case for assortative networks. In this parameterization, countries that engage in lots of incidents are likely to appear centrally in the latent space. We next consider a *negative slope* model by constraining $\theta_1 < 0$ in (3) so that two countries with lots of incidents between them are pushed further apart in the latent space. Under this parameterization, countries that engage in lots of incidents are likely to appear on the periphery of the latent space.

**Findings and Discussion**   We show the 2-D latent space plot with both positive and negative slope terms in Figure 2. We first consider the latent positions from the positive slope model. Notice that the most active nodes tend to appear centrally, and the node pairs with the most frequent incidents tend to be placed close together. For example, Israel (ISR) and Lebanon (LEB) have latent positions very close together, which makes sense given that they have the most incidents in the data set: 588 total incidents. Additionally, countries that are geographically close do mostly appear close together in the latent space. This can be seen from Figure 3, where nodes are colored by continent. The estimated parameters are $\theta_1 = 1.2, \theta_2 = -9.3, \alpha_1 = 0.77, \alpha_2 = 0.13$. The high value for $\alpha_1$ compared to $\alpha_2$ indicates the importance of self excitation in addition to reciprocal excitation.

Next, we consider the negative slope model. From examining the latent positions, we find that most active nodes tend to appear on the periphery of the latent space, which is reasonable because the model attempts to push nodes with many incidents far apart. For example, Israel and Lebanon are on opposite sides of the latent space. The estimated parameters for this model are $\theta_1 = -0.008, \theta_2 = -1.24, \alpha_1 = 0.83, \alpha_2 = 0.15$. While the parameters used for modeling the baseline intensity have changed significantly, the $\alpha$ parameters modeling self and reciprocal excitation are very similar to the positive slope model.

Additional results are presented in Appendix C. We note that this case study is intended to be exploratory rather than con-

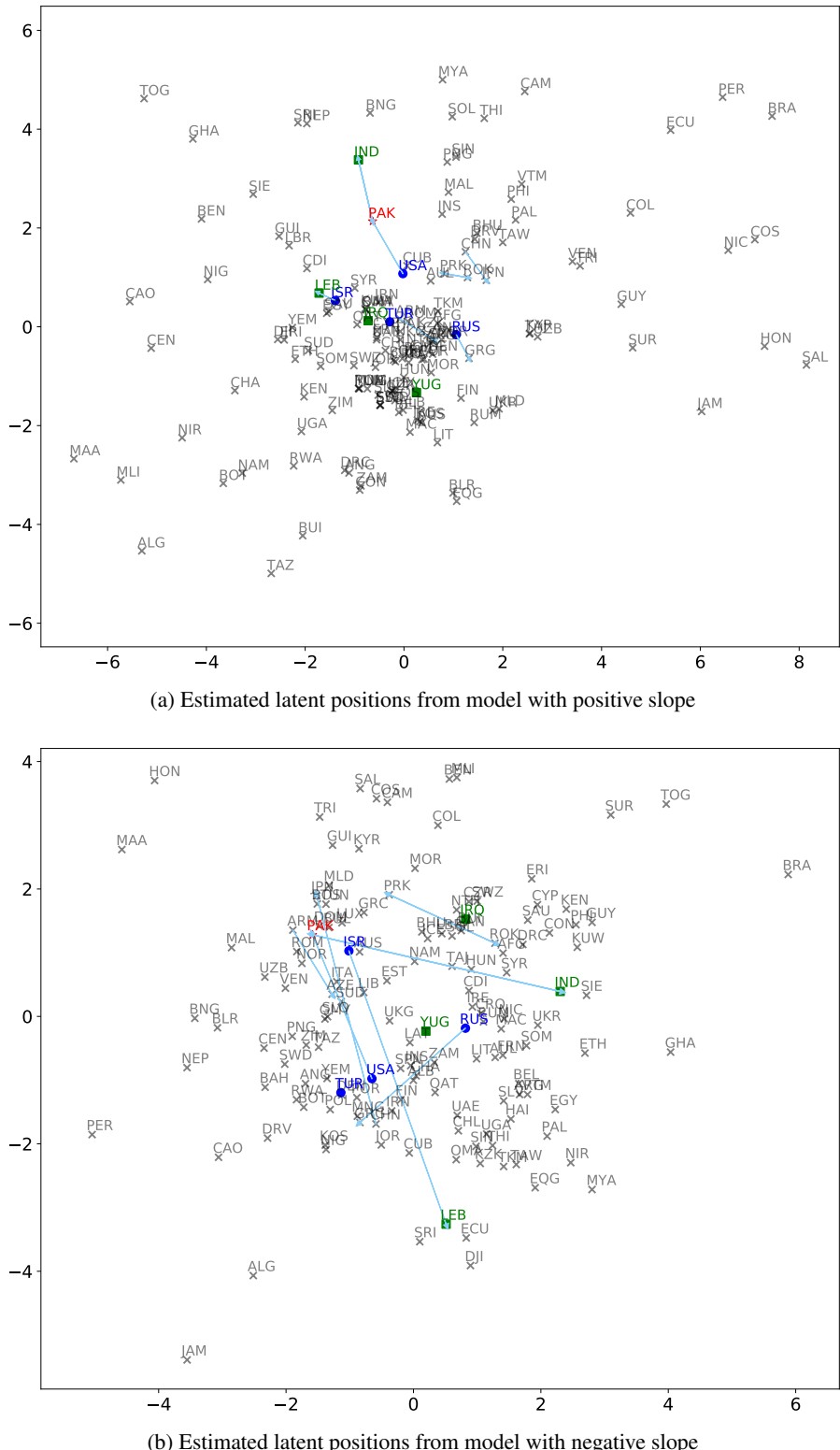

(a) Estimated latent positions from model with positive slope

(b) Estimated latent positions from model with negative slope

Figure 2: 2-D latent space plots for LSH model fit to MID data. Edges are shown for the 10 most frequently occurring incidents. The most active countries that initiate and receive the 5 most incidents are shown in blue and green, respectively. Pakistan (PAK) is among the top 5 initiators and receivers and is shown in red. (a) The model with positive slope places countries with lots of conflicts close together. The most active countries tend to appear centrally in this latent space. A zoomed in version of the center of the latent space is shown in Figure C.1 in Appendix C. (b) The model with negative slope places countries with lots of conflicts far apart. The most active countries tend to appear on the periphery of this latent space.

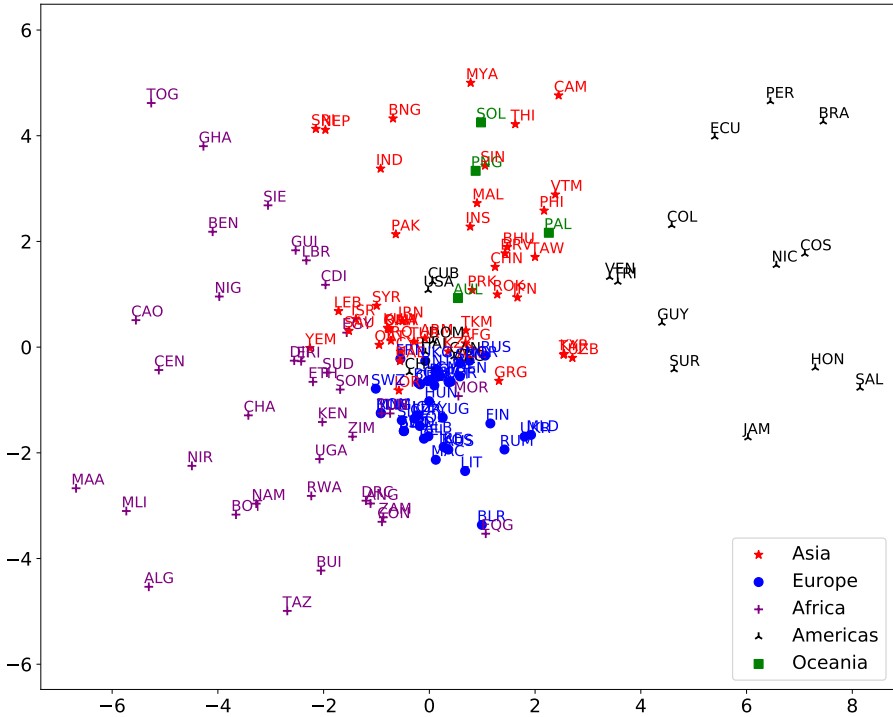

Figure 3: 2-D latent space plot for MID data with positive slope and countries colored by continent. A zoomed in version of the center of the latent space is shown in Figure C.2 in Appendix C.

firmatory. We caution readers from jumping to conclusions about countries from our results.

## 7 CONCLUSION

We proposed the latent space Hawkes (LSH) model for continuous-time networks of relational events, which models interactions between each pair of nodes as realizations from a mutually exciting Hawkes processes whose intensity functions include a baseline rate along with both self and reciprocal excitation terms. The LSH model makes use of a single latent space along with sender and receiver effects to provide a more interpretable fit while remaining competitive in accuracy compared to the dual latent space (DLS) model. We performed an exploratory analysis of militarized disputes between countries using the LSH, where the latent space was quite informative of the dispute network structure. We also found that self excitation was stronger than reciprocal excitation in this network, demonstrating the importance of self excitation, which is not present in the DLS model. We hope this paper inspires future work combining continuous latent space representations with TPPs, which have not gotten as much attention as block model-based TPPs.

**Limitations** While our proposed model shows superior empirical performance and interpretability, there are also several limitations. We use a single reciprocal jump size $\alpha_2$ for all node pairs, which results in a less flexible model

compared to the DLS, but it is more stable. While our estimation procedure scales to networks with about $1,000$ nodes, it does not scale to extremely large networks with $> 10,000$ nodes, unlike the the CHIP [Arastuie et al., 2020] and MULCH [Soliman et al., 2022] latent block models. Additionally, the latent positions of nodes in our LSH model are fixed over time, just like in the DLS. If there are significant changes in the network structure over time, a more flexible model that allows latent positions to change over time, such as the model of Rastelli and Corneli [2021], may be a better fit. Finally, one could model more complex dependencies among the nodes that goes beyond self and reciprocal excitation using a multivariate Hawkes process, as in the MULCH latent block model [Soliman et al., 2022], instead of a bivariate Hawkes process.

**Author Contributions**

Z. Huang, S. Paul, and K. S. Xu contributed to the model and algorithm development. Z. Huang and H. Soliman wrote the code and developed the experiments. All authors contributed to writing the paper.

**Acknowledgements**

This material is based upon work supported by the National Science Foundation grants IIS-1755824, DMS-1830412, IIS-2047955, and DMS-1830547.

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
