# OpenReview forum: "A Mutually Exciting Latent Space Hawkes Process Model for Continuous-time Networks"
_auai.org/UAI/2022/Conference — UAI 2022 Poster_

### Official Review · Reviewer_kEQa · 2022-04-07

**Q2(1) Originality/Novelty:** 3
**Q2(2) Significance/Impact:** 3
**Q2(3) Correctness/Technical Quality:** 3
**Q2(6) Clarity Of Writing:** 4
**Q6 Overall Score:** 7
**Q8 Confidence In Your Score:** 4

**Q1 Summary And Contributions:**

Paper proposes a new combination of a Hawkes process with an embedding for the types (network nodes in this case).  Experiments compare with prior embedding-based models of continuous-time network traffic data.

**Q2 Assessment Of The Paper:**

More detailed information regarding each of these aspects is given below:

**Q2(4) Quality Of Experiments (Optional):**

3: Good: The experimental evaluation is adequate, and the results convincingly support the main claims.

**Q2(5) Reproducibility:**

3: Good: Key resources (e.g., proofs, code, data) are available and key details (e.g., proofs, experimental setup) are sufficiently well-described for competent researchers to confidently reproduce the main results.

**Q3 Main Strengths:**

+ The model is a reasonable network embedding model that is motivated by both computational considerations and network theory.
+ The experiments are generally well carried out and explained (given space constraints).  They are convincing of the value of the approach.
+ The paper is well written
+ The model could be implemented by someone familiar with Hawkes processes without too much trouble.

**Q4 Main Weakness:**

- Identifiability claim is hard to follow and does not appear complete.  The theorem only demonstrates that a particular expression is identifiable.  Relating this back to the entire process and *data* is missing.
- Alternating optimization not sufficiently justified.  Why does this help?
- Method for performing prediction (or indeed any inference with the model) is not specified.

**Q5 Detailed Comments To The Authors:**

While this paper concentrates on embedding models for networks given continuous-time communication patterns, the work on probabilistic models for continuous-time communications goes back further than what is cited.  For instance, Snjiders (see below) had a latent-space model of dynamic networks in 2005.  While the latent space is not quite the same as the embedding here, there are certainly similarities.  The paper could be strengthened with some more complete literature review to orient this work with respect to other approaches.  Note that while many of these models are Markovian (or hidden Markovian), a HP with exponential kernel (or even a sum of exp) can be viewed as a Markov-state-space model.

HPs are often estimated using EM (the hidden variable is the previous event that "caused" any particular event).  Is there a reason this method was not used?  More directly, while the poor convergence of a straight L-BFGS is a suitable reason to search for an alternative method, it isn't clear why this alternating used of L-BFGS works better.  It seems like there is probably an important lesson to be learned here (and more general that this model).  Separately, calling "s_\Theta" and "s_Z" step sizes is misleading.  They are the *number* of steps taken by L-BFGS.

The appendix material suggests that prediction (for Section 5.2.1) is done via forward sampling.  This should be explicitly stated (along with how many samples are used).

Figure 2 is interesting, but should be improved in presentation.  The dots overlap with the text and the text is smaller than necessary.  These couple of changes would greatly enhance the figure's value, especially to the casual reader.

Citations to the datasets should be in the main text (even if the details are in the appendix).


ref:
Snijders, T. A. (2005). Models for longitudinal network data, chapter 11. New York: Cambridge Univ. Press.

Snijders, T. A., Steglich, C. E., & Schweinberger, M. (2007). Modeling the co-evolution of networks and behavior. In Longitudinal models in the behavioral and related sciences, chapter 4. Lawrence Erlbaum.

Wasserman, S. (1979). A stochastic model for directed graphs with transition rates determined by reciprocity. In K. Schuessler (Ed.), Sociological methodology, 392–412. Jossy-Bass.

Wasserman, S. (1980). Analyzing social networks as stochastic processes. J. Am. Stat. Assn., 75, 280–294.

Inference for these models was explored in UAI:
Yu Fan and Christian R. Shelton (2009). "Learning Continuous-Time Social Network Dynamics." UAI



**Q7 Justification For Your Score:**

The model is a well-considered one and probably the one I'd use were I to model the same.  The experimental results are convincing (at least in comparison to other recent embedding-based models).  The algorithms and theory behind the model are not novel (nor does the paper claim them to be).  In general, a solid paper describing a useful technique for modeling a particular (but common) type of network data.

**Q9 Complying With Reviewing Instructions:**

1: Yes.

---

### Official Review · Reviewer_tBvP · 2022-04-12

**Q2(1) Originality/Novelty:** 3
**Q2(2) Significance/Impact:** 3
**Q2(3) Correctness/Technical Quality:** 3
**Q2(6) Clarity Of Writing:** 4
**Q6 Overall Score:** 6
**Q8 Confidence In Your Score:** 4

**Q1 Summary And Contributions:**

The paper proposes a latent space hawkes process model for dynamic network embedding. The proposed one is an extended version of [Yang 2017] to make it more stable and also included self-excitation and node-specific effect in the model. Both simulated and real datasets have been used for performance comparison and superior results have been shown.

**Q2 Assessment Of The Paper:**

More detailed information regarding each of these aspects is given below:

**Q2(4) Quality Of Experiments (Optional):**

3: Good: The experimental evaluation is adequate, and the results convincingly support the main claims.

**Q2(5) Reproducibility:**

3: Good: Key resources (e.g., proofs, code, data) are available and key details (e.g., proofs, experimental setup) are sufficiently well-described for competent researchers to confidently reproduce the main results.

**Q3 Main Strengths:**

Dynamic network embedding by integrating point process like Hawkes process to model events happening over continuous time is an interesting and important problem, even though it has been studied for some years.
This paper is well-written and clearly organized, with implementation details clearly provided.
The proposed method is a reasonable one, and adding back the self-exciting and node-specific effect are reasonable ones.
The experiments are carefully carried out and clearly presented.

**Q4 Main Weakness:**

While adding back the self-exciting and node-specific effect are reasonable ones, they are not particularly very new ideas. Also, I consider this work to be an extension of the existing work [Yang 2017]. Thus the originality of this work is discounted.



**Q5 Detailed Comments To The Authors:**

1) The case study was only performed on one dataset. It will be good to do that also for the other datasets so as to better understand its applicability in general.

2) Limitations of the proposed model are not much discussed.

3) For an event between two nodes, I suppose there can be direction? How can they be considered in the proposed model?

4) Can the proposed model be applied to big networks with say hundred of thousand nodes? Related discussion could be useful.

5) It seems that the formulation assumes an event can only happen between two entities. In general, it could involve multiple entities. Could such cases be addressed as well?



**Q7 Justification For Your Score:**

The paper is well-presented and logically well-articulated. The proposed extensions are reasonable, and carefully formulated and implemented. The evaluation is also carefully conducted. Yet, I do not consider the novelty to be a very significant one.

**Q9 Complying With Reviewing Instructions:**

1: Yes.

---

### Official Review · Reviewer_7B63 · 2022-04-12

**Q2(1) Originality/Novelty:** 3
**Q2(2) Significance/Impact:** 3
**Q2(3) Correctness/Technical Quality:** 3
**Q2(6) Clarity Of Writing:** 4
**Q6 Overall Score:** 6
**Q8 Confidence In Your Score:** 4

**Q1 Summary And Contributions:**

This paper proposes a new latent space Hawkes process to model continuous-time networks, where a special conditional intensity function is designed to capture both self-exciting and mutually exciting. Such simple design has achieved relatively good performance on various tasks.

**Q2 Assessment Of The Paper:**

More detailed information regarding each of these aspects is given below:

**Q2(4) Quality Of Experiments (Optional):**

3: Good: The experimental evaluation is adequate, and the results convincingly support the main claims.

**Q2(5) Reproducibility:**

3: Good: Key resources (e.g., proofs, code, data) are available and key details (e.g., proofs, experimental setup) are sufficiently well-described for competent researchers to confidently reproduce the main results.

**Q3 Main Strengths:**

1.	The problem is well-motivated, and the literature is well explained and discussed;
2.	The idea is clearly explained, and it is easy to see the advantages of the new design;
3.	Sufficient experiments are conducted to verify the effectiveness of the proposed new model;
4.	The experimental results are promising.


**Q4 Main Weakness:**

The main and only contribution of this work lies inthe design of the new conditional intensity function, but it looks a little straightforward. The other parts, like estimation, just follow the literature.

**Q5 Detailed Comments To The Authors:**

The main and only contribution of this work lies on the design of new conditional intensity function, but it looks straightforward. Are there any other potential designs? How to converge to the current one? What is the additional benefit from such design comparing with other potential ones?

When comparing with DLS, the authors claim that the DLS is not stable. Can you please talk more on that? If the possible >1 jump size is the reason, why does not happen for the alpha in Eq. (2)? I think the scale of alpha \exp{||x_u - x_v||^2} can be controlled as well as the alpha in Eq. (2).

In the proof of Theorem 3.1, why can you get n^2(\theta_1 - \theta_1\prime) = 0 after right multiplying by 1_n? because n(\eta - \eta^\prime)1_n = 0? Then, why?


**Q7 Justification For Your Score:**

Overall, this is a good paper with clear motivation, idea, writing, sufficient experiments, and details. My only conerns arethe relatively weak contribution and some minor questions.

**Q9 Complying With Reviewing Instructions:**

1: Yes.

---

### Official Review · Reviewer_WgXP · 2022-04-13

**Q2(1) Originality/Novelty:** 2
**Q2(2) Significance/Impact:** 2
**Q2(3) Correctness/Technical Quality:** 3
**Q2(6) Clarity Of Writing:** 3
**Q6 Overall Score:** 5
**Q8 Confidence In Your Score:** 3

**Q1 Summary And Contributions:**

This paper proposes a latent space Hawkes (LSH) model for relational event data, which uses a latent space representation in mutually exciting Hawkes processes where the baseline intensities depend on distances between nodes in latent space. An alternating minimization algorithm is proposed to estimate latent positions of nodes and other model parameters. The main contribution is an extension of Hoff et al. 2002, and seems to be a simplification of prior work such as DLS (Yang et al. 2017).

**Q2 Assessment Of The Paper:**

More detailed information regarding each of these aspects is given below:

**Q2(4) Quality Of Experiments (Optional):**

3: Good: The experimental evaluation is adequate, and the results convincingly support the main claims.

**Q2(5) Reproducibility:**

3: Good: Key resources (e.g., proofs, code, data) are available and key details (e.g., proofs, experimental setup) are sufficiently well-described for competent researchers to confidently reproduce the main results.

**Q3 Main Strengths:**

The main strength of the paper is that it shows some performance advantages over closest work in the space, e.g. DLS (See Table 1). The authors argue that their simpler method has different types of benefits. They also discuss aspects such as generative accuracy, interpretability about the latent space, etc.

**Q4 Main Weakness:**

The main weakness of the paper is that the technical contributions seem somewhat incremental to me, over closest prior work such as DLS. There appears to be a lot of literature for relational event models, and it is unclear to me what impact this paper could have over this research space.

**Q5 Detailed Comments To The Authors:**

I feel the introduction could be more informative, particularly about the motivation behind the work. There is clearly a lot of literature in this space, so what was the motivation behind this direction? It looks like some content from p4 about comparison with DLS could potentially be leveraged in the introduction.

A very specific approach for the Hawkes process is used -- a sum of exponential kernels method. Perhaps it might be helpful for more justification around this choice. And did the authors try other kernels?

I did not follow some parts of the identifiability section on p3. Perhaps a summary would help? As I understand, theta_1 is not identifiable but most other parameters are identifiable. What about Z?

The authors say they are tackling the prediction task on p5 but this feels more like a model fitting task, as they use log-likelihood to gauge model fit on test data.

I found the generative accuracy section on p6 to be novel, i.e. different from other work in this space. However, I feel I didn’t follow this entirely, perhaps because I don’t understand DLS. How would generative accuracy be different from predictions, i.e. when future events happen given the past?

What is a disassortative network (mentioned on p7)?

On p7, it is unclear which is the more appropriate model for the MID case study – positive or negative slope? Could the authors say more about how their model would be used in practice?

I appreciate the case study but the figure on p8 did not really provide me with much insight – and it takes up a lot of space! Could the authors think of a more compact way to explain the latent space idea here? What is the reader expected to take away here? This figure could definitely use some modifications.

**Q7 Justification For Your Score:**

I feel the paper has both strengths and weaknesses which balance each other. However, I note that I do not know the closest literature well, so I will rely on comments from other reviewers during discussion.

**Q9 Complying With Reviewing Instructions:**

1: Yes.

---

### Decision · Program_Chairs · 2022-05-15

**Decision:**

Accept (Poster)

**Comment:**

Meta Review: This paper proposes a latent space Hawkes process to model continuous-time networks. The reviews all agree that this paper is well presented, has clear motivations, and has sufficient empirical evaluations. One concern is that the underlying idea is incremental.